

# An activity theory-based exploration of "Eyeland", a task-based serious game for EFL visually impaired students

Karen Villalba[1], Heydy Robles[1], Miguel Jimeno[2], Martha Cecilia Delgado-Cañas[3], Adriana Perez[4] and Francisco Quintero[5]

[1] Foreign Languages Department, Universidad del Norte, Barranquilla, Atlantico, Colombia
[2] Computer Science Department, Universidad del Norte, Barranquilla, Atlantico, Colombia
[3] Education Operations Department, St George University of London, London, Greater London, United Kingdom
[4] Spanish Department, Universidad del Norte, Barranquilla, Atlantico, Colombia
[5] Psychology Department, Universidad del Norte, Barranquilla, Atlantico, Colombia

Corresponding author
Karen Villalba,
karen.villalba.ramos@gmail.com

## ABSTRACT

This study investigates how the Eyeland app, an accessible task-based serious game for English as a foreign language (EFL), can remediate traditional lessons for both visually impaired students (VISs) and sighted students (those without visual impairments) in a public high school in Colombia. Using an activity theory framework and its derived model, the Activity Theory-Based Model of Serious Games (ATMSG), the study explores the characteristics of traditional EFL lessons designed for these students, the adjustments made while integrating the Eyeland app, and the resulting changes in student experiences. The research employed action research cycles involving teachers in reflective processes that included planning, observing, acting, and re-planning to adapt and integrate Eyeland into the classroom. Qualitative research methods—field observations, focus groups, usability surveys, teacher interviews, and document analysis—were used to collect data and analyze how Eyeland was implemented and its effects on teaching and learning. Findings indicate that Eyeland effectively remediated traditional lessons by offering accessible, interactive features such as auditory, tactile, and visual support. These enhancements improved engagement for both sighted and visually impaired students. Traditional EFL lessons, which relied heavily on visual materials and teacher-centered methods, were transformed into more interactive, task-based activities that encouraged greater collaboration and student autonomy. Adjustments included redesigning lesson plans and rearranging classroom layouts to foster inclusion. Both sighted and visually impaired students reported positive experiences, particularly valuing the increased autonomy and engagement provided by Eyeland's interactive tasks. The study highlights significant changes made to the lessons, with visually impaired students reporting predominantly positive experiences, though some teachers were reluctant to engage with the app. Eyeland contributed to the creation of inclusive classrooms by shifting from a teacher-centered approach to a student-centered learning environment that promoted better language learning outcomes. ATMSG was instrumental in analyzing how Eyeland fostered inclusive learning practices and provided valuable insights into re-mediation strategies, pedagogical planning, and the development of accessible content for EFL learners.

## INTRODUCTION

A growing paradigm in education involves the incorporation of mobile-assisted language learning (MALL) into conventional pedagogical practices. However, the alignment of mobile technology with learning theories, especially for diverse learners such as visually impaired students (VISs), remains underexplored. Despite the increasing integration of technology in classrooms, several critical elements—such as the learner, the learning context, and the role of tutors—are often overlooked in mobile learning applications (*Grant, 2019*; *Koole, 2009*; *Krotov, 2015*; *Mouza & Barrett-Greenly, 2015*). Furthermore, technical and accessibility limitations, insufficient experience with technology, the prohibition of cell phones in schools, and a lack of curricular adaptations have created barriers to fully realizing inclusive classrooms (*Seprilia, Handayani & Pinem, 2017*). To address these gaps, Activity Theory offers a vital framework for analyzing how mobile learning tools interact with these overlooked elements, particularly for students with disabilities.

For visually impaired students, MALL does not automatically ensure the support needed for their learning process (*Corbeil & Valdes-Corbeil, 2007*; *Khan, Nazir & Khan, 2021*). While MALL offers the potential for greater accessibility, there is still a lack of specific policies and frameworks for integrating accessible apps into English as a foreign language (EFL) classrooms. For instance, *Senjam, Manna & Bascaran (2021)* found that despite the availability of mobile devices, VISs face significant challenges in accessing them for learning purposes. These challenges highlight the need for critical disability theory, emphasizing the importance of inclusive teacher education and socially equitable pedagogy (*Burciaga & Kohli, 2018*; *Hosking, 2008*; *Philip et al., 2019*). This perspective stresses the importance of designing technology that accounts for the diverse needs of learners, particularly those with disabilities, and ensures equitable access to educational opportunities.

Activity Theory provides a robust framework for exploring how VISs interact with MALL tools, particularly in task-based learning environments like Eyeland, the task-based serious game explored in this study. By focusing on the activities and contexts in which learning occurs, educators can design more effective and inclusive learning experiences (*Cohen & Ezra, 2018*). Studies have shown that VISs can significantly improve their language skills and overall learning experiences when provided with accessible MALL tools (*Retorta & Cristovão, 2017*). The current research builds on this by applying Activity Theory to analyze how the Eyeland app scaffolds learning for VISs, thereby promoting both autonomy and collaboration within EFL classrooms.

Existing studies on MALL have provided valuable insights into various aspects, including the use of devices, memory effects, students' attitudes and perceptions, and their autonomy and self-regulation (*Lai & Zheng, 2018*; *Lyddon, 2016*; *Xu & Peng, 2017*).

Research has shown that the use of mobile devices in language learning can significantly enhance students' ability to recall target vocabulary, resulting in improved performance in language tasks (*Sato, Murase & Burden, 2020*). Moreover, students tend to have positive perceptions of MALL, particularly appreciating the features of gamification, ease of use, and the ubiquitous nature of mobile applications, all of which contribute to a more engaging learning experience and heightened motivation (*Gafni, Achituv & Rahmani, 2017*; *Nuraeni et al., 2020*).

Additionally, studies suggest that MALL fosters an increased capacity for self-regulation among learners, a key factor in effective language acquisition (*Viberg, Kukulska-Hulme & Peeters, 2023*). By offering flexible learning environments and personalized content, it supports learner autonomy, enabling students to engage with learning materials at their own pace and according to their individual preferences, an essential aspect of successful language learning (*Lei et al., 2022*).

While these are important findings, they often overlook the specific challenges faced by VISs in MALL environments. Additionally, mobile learning tools in language education are often used as supplementary resources for practice rather than being fully integrated into classroom tasks, limiting their effectiveness (*Burston, 2014*; *Purgina, Mozgovoy & Blake, 2020*).

Regarding inclusion, collaborative learning strategies that take a learner-centered approach to encourage social interaction and prevent marginalization have gained importance (*Crompton, 2013*; *Lan, Sung & Chang, 2007*; *Lin & Yunus, 2012*). These approaches align with personalized language learning through mobile apps, promoting student autonomy (*Çakmak, 2019*; *Hoven & Palalas, 2011*). However, research continues to call for the redesign of mobile learning environments to meet the needs of diverse learners, especially students with disabilities. One promising avenue for achieving this is the use of serious games, such as Eyeland, which have been shown to support language learning by combining various skills and approaches in an engaging, goal-oriented format (*Chen & Hsu, 2020*).

By addressing the gap in the literature regarding VIS, this research provides guidelines for incorporating accessible EFL serious games into regular lessons; emphasizing the importance of assistive technologies and their connections to universal design concepts to guarantee access to education (*Alajarmeh, 2022*; *Leria, Benitez & Fraga, 2021*); and determining the accessibility, gamification, and mobile language learning aspects of Eyeland from a technological and pedagogical perspective through the support of Activity Theory to analyze the app as a tool and its effect in scaffolding both teachers' lesson planning and students' learning.

The current investigation is situated within the specific context of Colombian public schools. The study is conducted within an institution that endeavors to foster inclusivity in a vulnerable setting characterized by certain limitations, particularly in its location within an urban-marginal area. The following research questions have been formulated to provide a structured framework for guiding this research:

1) How can an EFL-accessible task-based serious game app (Eyeland) remediate traditional lessons for both sighted and visually impaired students at a public high school in Colombia?

2) What are the characteristics of traditional EFL lessons designed for both sighted and visually impaired students?

3) What adjustments were made to the EFL lessons while the Eyeland app was used for both sighted and visually impaired students?

4) What are the experiences of sighted and visually impaired students using Eyeland?

5) What activity re-mediation occurred while using Eyeland?

## Literature review

This review focuses on examining studies related to applications for VISs, assistive technology, and mobile task-based learning. It explores how accessibility and inclusion are incorporated into the design of serious games and assesses their impact on EFL pedagogy for VISs.

## Apps for VISs

The use of mobile devices for supporting learning through screen-based technology and interaction has gained substantial attention in recent years. Specifically, MALL is recognized as a versatile and adaptive approach that leverages personal electronic devices to enhance learning in various contexts, with a strong emphasis on fostering interactivity, promoting individualized learning experiences, and improving context awareness (*De Vega, Basri & Nur, 2023*; *Pérez-Paredes & Zhang, 2022*). However, VISs face additional challenges in achieving digital literacy and full inclusion in such digital learning environments (*Akcil, 2018*; *Kamaghe, Luhanga & Kisangiri, 2020*; *Trongtortam, 2019*).

Mobile technology has significantly influenced the education sector for over a decade, with *Fajaruddin et al. (2024)* highlighting how mobile devices align with the skills needed for 21st-century learning. Furthermore, MALL has been shown to increase motivation and engagement among students, both inside and outside the classroom (*Shaheen, Soomro & Ali, 2024*). The increasing adoption of mobile learning in schools can be attributed to its ease of use and the incorporation of gamification elements, which help create a more interactive and playful learning environment for students (*Kao, Yuan & Wang, 2023*).

A review by *Torres-Carazo, Rodríguez-Fórtiz & Hurtado (2016)* highlighted that categories such as education, entertainment, and games offer the most suitable apps for VISs due to their customizable accessibility options. Similarly, *Griffin-Shirley et al. (2017)* found that 95.4% of participants considered apps beneficial for VISs due to their accessibility features. Nonetheless, challenges persist in addressing braille interfaces, writing and spelling challenges, and screen reader optimization (*Al-Razgan et al., 2021*; *Damaceno, Braga & Mena-Chalco, 2018*; *Jaramillo-Alcázar & Luján-Mora, 2017*; *Kamali Arslantas, Yıldırım & Altunay Arslantekin, 2019*). Studies such as *Karthika & Selvam (2022)*, which explored teaching English to blind students using WhatsApp voice notes, and *Nahar, Sulaiman & Jaafar (2021)*, which developed a Bangla Braille learning

application for VISs, underscore the potential of mobile technologies to foster language learning in visually impaired populations.

## Assistive technology and mobile learning

In the field of education, assistive technology refers to products and practices designed to enhance the learning potential of both typical students and those with special educational needs or learning difficulties. These technologies aim to mitigate the impact of potential barriers that may impede learning (*Yenduri et al., 2023*). As cutting-edge tools, assistive technologies significantly enhance students' abilities, allowing them to complete academic tasks more efficiently and effectively (*O'Sullivan et al., 2023*). Research indicates that both students with learning disabilities and those who struggle with comprehension strategies can benefit from the use of assistive technology (*Al-Dababneh & Al-Zboon, 2022*). Mobile learning, in particular, presents new possibilities for improving the teaching and learning process for students with disabilities studying foreign languages (*Svalina & Ivić, 2020*). Providing clear and contextualized instructions is essential for students facing learning challenges (*Miller, 2016*). This is supported by *Bunch-Crump & Lo (2017)*, who utilized computer-assisted morphological instruction to assist English learners with reading impairments. While assistive technology, when combined with mobile learning tools, has the potential to enhance language acquisition, further research is necessary to determine how these tools can be tailored to the specific needs of VISs.

Research shows that both students with learning disabilities and those who lack comprehension methods can benefit from assistive technology (*Al-Dababneh & Al-Zboon, 2022*). Mobile learning offers new opportunities for enhancing the teaching and learning process for students with disabilities studying foreign languages (*Svalina & Ivić, 2020*). Clear and contextualized instructions are crucial for students with learning difficulties (*Miller, 2016*), as demonstrated by *Bunch-Crump & Lo (2017)*, who used computer-assisted morphological instructions to aid English learners with reading impairments. Assistive technology combined with mobile learning tools can enhance language acquisition, but more research is needed to explore how these tools can be tailored to meet the specific needs of VISs.

## Mobile task-based learning

Mobile-supported task-based learning has shown promise in improving students' language performance by facilitating the integration of linguistic skills and promoting a student-centered approach (*Aliasin, Saeedi & Pineh, 2019*; *Ellis, 2012*; *Nakahama, Tyler & van Lier, 2001*; *Xue & Churchill, 2019*). These mobile tasks, which enhance psychological factors such as engagement, autonomy, and interest, have proven more effective than traditional task-based lessons (*Fang et al., 2021*; *Jia & Harji, 2022*; *Tragant et al., 2022*). Despite these advancements, there remains a need for mobile-enhanced lesson planning that incorporates accessibility features, particularly for VISs (*Hockly, 2016*).

## Lesson planning for VIS and MALL

The ability to incorporate accessibility elements into MALL lesson planning is crucial for success (*Pegrum, 2014*). Teacher training and lesson design must address the varied levels of students, including those with disabilities, to implement MALL effectively (*Burden, Hopkins & Pike, 2011*). Recent studies emphasize the importance of mobile-enhanced lesson plans that accommodate these diverse needs (*Luís, 2016*), with inquiry-based learning as a valuable approach (*Goldston et al., 2013*). This method fosters active learning, where students engage in real-world tasks to deepen their knowledge (*Lameras et al., 2021*). For VISs, these pedagogical models must integrate mobile devices to promote autonomous learning while maintaining accessibility.

For VISs, these pedagogical models must integrate mobile devices to promote autonomous learning while ensuring accessibility. The inquiry-based learning model aligns well with task-based learning and emphasizes immersive and situated learning experiences, providing opportunities for students to apply their knowledge in practical settings (*Agustin et al., 2015*).

The 5E model serves as an instructional framework for inquiry-based learning, comprising five phases: engagement, exploration, explanation, elaboration, and evaluation (*Bybee et al., 2006*). During the engagement phase, the teacher assesses prior understanding and generates interest. The exploration phase involves hands-on activities and collaborative work, followed by the explanation phase, where students articulate their ideas and receive concept labels and definitions. The elaboration phase encourages the application of skills in real-world contexts, and the evaluation phase includes a summative assessment aligned with the inquiry objectives (*Schallert, Lavicza & Vandervieren, 2022*).

## Serious games and visual impairment

Serious games which are defined as structured games designed for educational purposes, offer significant potential for improving EFL learning for VISs (*Cheng et al., 2017*; *Krath, Schürmann & Von Korflesch, 2021*; *Qian & Clark, 2016*). Some studies have explored how serious games can facilitate the process of acquisition and retention of new vocabulary, as has been described in terms of perceptions by Duolingo users whose flexibility and gamification aspects are positive (*Loewen et al., 2019*). Other works suggest that serious games can enhance vocabulary acquisition and recall (*Sandberg, Maris & Hoogendoorn, 2014*; *Franciosi, 2017*). A review of several games, such as AudiOdyssey, Crystal Island, and Blind Legend, underscores the importance of accessibility features like multiple input controls, audio cues, and touch-based interfaces for VISs (*Csapó et al., 2015*; *De Sousa, da Silva Júnior & Ferreira, 2021*; *Glinert & Wyse, 2007*). These games, which use auditory and tactile feedback mechanisms, provide immersive learning experiences that allow students to navigate educational content with ease (*Israel, Wang & Marino, 2016*; *Taub et al., 2020*). However, there is a need for further research on how serious games can be adapted for use in EFL classrooms for VISs, particularly in the Global South (*Effendi, Thurston & MacKenzie, 2024*).

Although significant progress has been made in developing mobile learning tools and serious games for EFL learners, gaps remain in addressing the specific needs of VISs.

Recent studies (*Cezarotto, Martinez & Chamberlin, 2022*; *Othman, Mohamed & Mat Zin, 2023*) have underscored the importance of designing serious games that integrate accessibility, game design, and pedagogical components, yet few studies focus specifically on VISs in EFL contexts. Additionally, the creation of supportive learning environments that incorporate peer interaction and social support has been recognized as critical for the success of VISs in language learning (*Nagar & Krisi, 2023*; *Rahman et al., 2024*). Despite advancements in mobile learning and serious games, challenges such as individualized support and varied levels of vision impairment persist. This research aims to fill this gap by exploring how mobile-assisted serious games can be tailored to enhance the learning experiences of VISs, focusing on accessibility, task-based learning, and pedagogical strategies that promote inclusion.

## Activity theory

An activity theory framework is used to determine how an accessible EFL serious game app (Eyeland) allows students with visual impairments to achieve A1-level language learning outcomes. This framework was selected since it incorporates strong notions of mediation (activities mediated by artifacts, both internal and external), as the central unit of analysis for human interaction and notions of computer–human interaction and development (*Bødker, 1996*; *Engeström, 1993*; *Uden, 2007*; *Gibbes & Carson, 2014*). In addition, it matches the objective of critical disability theory, in which people with disabilities want to free themselves from repressive social policies, practices, and stereotypes (*Gillies, 2014*), with what *Blunden (2010)* highlights from the theory: its emancipatory underpinnings. Finally, the theory is valorized for apprehending the dynamics of complex situations and for identifying contradictions (understood as drivers for change) (*Bligh & Flood, 2017*), which might frame action research design.

As shown in Fig. 1, the core components of the activity theory framework illustrate the relationship between various elements involved in human interaction. This model is particularly valuable for understanding how activities, especially in educational contexts, are mediated by tools and shaped by social structures.

The idea behind this theory is that all activities are mediated by culturally specialized tools. The concept of learning is redefined by an activity that is being mediated and has significant implications for mobile learning. Instead of learning to rationally abstract mental representations from one's personal experience, learning on the basis of activity theory is instead characterized as learning to participate in a cultural practice (*Gifford & Enyedy, 1999*). In this way, apps and technologies such as them take the role of traditional language-learning meditations, generating fresh actions on an individual level and working together to create a brand-new activity, possibly in unforeseen or unexpected ways (*Lektorsky, 2009*).

This means that we can develop new perspectives and remediate phenomena for diverse communicative goals. As we employ new mediating artifacts or technology, they help shape our behavior and thinking in qualitatively new ways (*Cole & Griffin, 1986*; *Nilsen et al., 2021*). This re-mediation will be analyzed from three different perspectives: accessibility features of apps validated by *Vendome et al. (2019)*; a serious game taxonomy

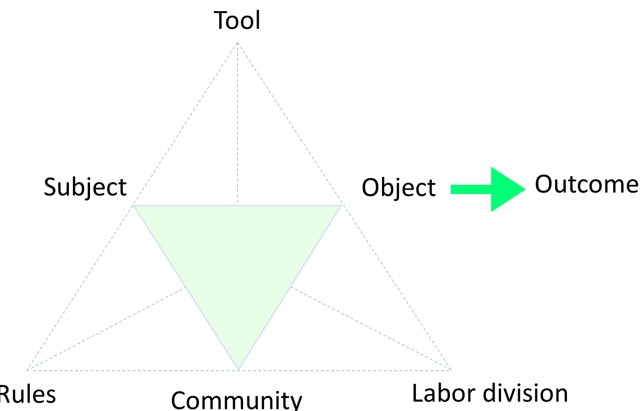

**Figure 1 The activity theory framework.** In activity theory, the basic unit of analysis of all human endeavors is activity: a purposeful interaction between a subject and an object in a process in which mutual transformations are accomplished. Model developed by *Engestrom (2000)* and refined by *Carvalho et al. (2015)*.

by *De Lope & Medina-Medina (2017)*; and a comprehensive framework for mobile language learning under the proposal of *Cacchione et al. (2015)*.

Building on this framework, the activity theory-based model of serious games (ATMSG) highlights the interconnections between game elements throughout gameplay and their role in achieving the intended pedagogical objectives. Rather than viewing the game as a standalone entity, this model recognizes it as an integral component of a complex and dynamic system involving learners and educators. This perspective facilitates a meticulous representation of serious educational games, elucidating the interconnectedness of game elements in contributing to pedagogical goals (*Callaghan et al., 2018*; *Carvalho et al., 2015*).

As illustrated in Fig. 2, the ATMSG model visually depicts how learning activities and gaming activities are integrated, highlighting their role in facilitating the subject's interaction with the object. This interaction ultimately contributes to the achievement of learning outcomes by aligning both the gaming and instructional components to the learner's motives.

Eyeland is described in terms of architecture design, components, and tasks and uses this very specific model of activity theory: ATMSG, which comprises gaming, learning and instructional activities for the particular situation of VISs and the learning scenario mentioned.

Figure 3 portrays the Eyeland activity framework, which details the architecture design, components, and tasks.

The framework proposed by *Sharples, Taylor & Vavoula (2005)* offers a versatile approach to investigating mobile technology as a learning tool through activity theory. It explores the interconnections among subjects, objects, communities, rules, division of labor, tools, and outcomes. It also examines how mobile technology facilitates blended learning, encompassing face-to-face and online activities, while considering the perspectives of sight and VIS. *Lin et al. (2019)* further analyzed the interconnections among these aspects, presenting a model that effectively represents the complex dynamics

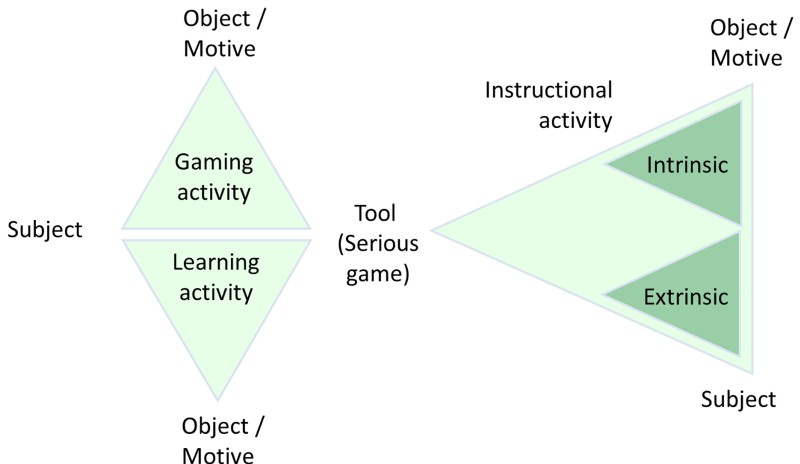

**Figure 2 The ATMSG model.** *Carvalho et al. (2015)* proposed the ATSMG model to show that engaging in serious games for educational purposes encompasses three primary activities. The depicted figure illustrates the elevated tier of the activity system, highlighting the three main activities and delineating the relationships among individuals and artifacts within this system.

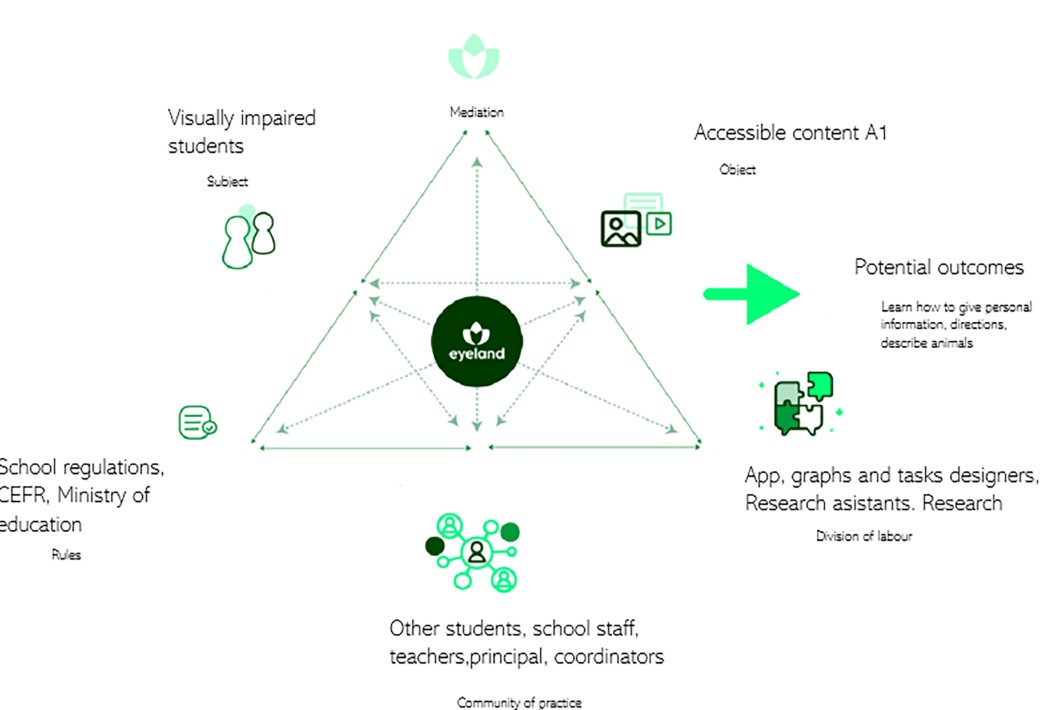

**Figure 3 The general system of Eyeland app.** The activity theory model serves to illustrate the object and its outcomes. The model was expanded to encompass not only multiple perspectives but also dialogs among various interacting systems, with Eyeland functioning as a significant artifact in this context (source prepared by the authors).

of activity theory and its effects on language skills. By considering diverse elements, including technology integration and the needs of VIS, the diagram will then offer a comprehensive overview of blended learning's multifaceted nature and teacher scaffolding.

## MATERIALS AND METHODS

### Research design and approach

For this study, a qualitative approach was selected since it constitutes the most appropriate way to interpret and understand social realities (*Hernández Sampieri et al., 2017*; *Martínez, 2004*; *Taylor & Bogdan, 1987*; *Valles, 2000*), particularly for people with visual impairment and blindness. In addition, according to *Glanz (2003*, p.10), this implies making "a detailed verbal description of the observed phenomena" by studying them in their natural contexts to give them a meaning or interpretation through a hermeneutical exercise (*Martínez, 2006a*, *2006b*).

To this end, the path followed for the development of this study was that of action research (*Kemmis & McTaggart, 2014*). As it is a cyclical process that involves planning, action, systematic observation and reflection, its purpose is to serve as a means to improve educational practices and solve problems in vulnerable educational contexts. Thus, a type of self-reflective understanding is generated so that participants understand why they are frustrated by the conditions under which they act and what they suggest changing (*Yuni & Urbano, 2006*). Therefore, this integration of qualitative methods allows for a richer and more contextualized understanding of the problems and solutions to facilitate the implementation of effective and sustainable changes in English teaching and learning in educational diversity. This to improve education through its change and learn through it.

In addition, action research is a participatory method that encourages self-reflection and continuous improvement through a spiral of four iterative cycles. To replan the action again, new observations and reflections, as long as the spiral practices processes of dialogue and criticism in the group and in each phase of the process. From this scenario, participants (teachers, students with and without disabilities) are involved collaboratively in an educational social situation, with the aim of improving language learning through technology.

Figure 4 shows the cycle of the research-action spiral. Each cycle began with a phase of preparation and reflective diagnosis to examine the problem situation, identify the needs to understand the context and anticipate the meaning of action with the formulation of hypotheses and interpretative and action objectives, for example, the difficulties and challenges of learning a foreign language with a disability. This was followed by the planning phase, in which teachers and researchers designed the lesson collaboratively using Eyeland and set specific learning objectives. This is based on open and exploratory research questions, and considering teaching experience, the characteristics of teaching praxis, and the review of literature and other textual sources that guide the teaching and learning of foreign languages in Colombian educational contexts, for example, Basic Learning Rights (DBA), the Basic Standards of Foreign Language Proficiency: English and the Curriculum suggested by the Ministry of National Education (https://www.mineducacion.gov.co/portal/).

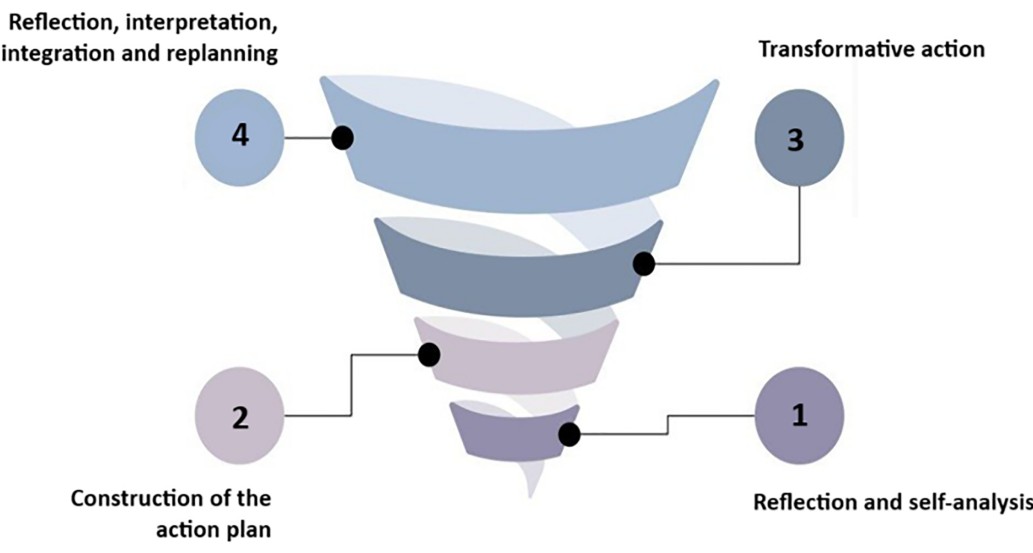

**Figure 4 Phases and moments of the methodological process.** The cycle of the research-action spiral is shown.

During the performance phase, the lesson was implemented in the classroom, and students used the app to participate in task-based language learning activities. All these actions of a flexible nature were adapted to respond to the emerging nature of the study, based on the resources used and the interactions, with the contents, with the technology and with the challenges.

The observation phase involved both real-time classroom observation and video recordings by a co-investigator to document students' interactions with the app and with each other. During this phase, the data is analyzed and the meanings and contexts behind the actions and experiences observed are sought to be understood and reflected. A valuable aspect in the study, since it sought to understand the experiences and perspectives from the point of view of the participants.

Finally, the reflection phase included briefings with teachers after the lesson, in which the effectiveness of the lesson was discussed, and adjustments were made to improve the subsequent cycle. This is based on the identification of emerging themes and categories for analysis, allowing adjustments and refinements in the intervention.

Both activity theory and action research, through their interpretive approach, can lead to critical reflections that can be documented to improve lesson planning under accessibility, gamification, and language learning parameters because the goal of this research is to determine how Eyeland can re-mediate traditional lessons for public high school VISs. To enhance teaching efficacy and assess the effects on the learning process, the action research cycles recommended in educational contexts are frequently used.

## Course design and learning activities

The course was designed around Eyeland, an EFL-accessible task-based serious game app, which was integrated into the curriculum to facilitate interactive learning experiences.

Lessons incorporated both individual and collaborative learning activities, where students—both sighted and visually impaired—engaged with the app to practice vocabulary, grammar, and conversational skills. Each session began with the teacher introducing the language objectives for the day, followed by a set of tasks in Eyeland that aligned with the lesson's objectives.

For example, students were tasked with completing missions or challenges within the game, which involved solving language problems using the app's accessible features. These tasks encouraged both autonomous learning and peer collaboration, as students had to interact with the content and each other to complete the activities. The use of gamification elements within Eyeland (*e.g.*, points, rewards, and levels) was designed to enhance student motivation and engagement.

### Ensuring reliability and validity

To ensure the reliability and validity of the data, triangulation was employed, combining multiple data sources (observations, focus groups, surveys, interviews, and document analysis) to corroborate findings and ensure a well-rounded understanding of the intervention. Furthermore, member checking was conducted during the reflection phase of the action research cycles, where teachers reviewed the preliminary findings and provided feedback to confirm the accuracy of the interpretations. Additionally, peer debriefing sessions were held with co-researchers to discuss the data and ensure consistent analysis.

### Data analysis techniques

Data were analyzed using an interpretive approach, informed by Activity Theory, which provided a framework for understanding how the introduction of Eyeland re-mediated the classroom activity system. Qualitative data from observations, focus groups, and interviews were transcribed, with a focus on identifying patterns in student engagement, accessibility issues, and the impact of the app on teaching practices. The data analysis process followed the steps of familiarization, coding, and generating themes to ensure a rigorous exploration of the findings. Special attention was given to the accessibility and gamification elements of Eyeland, as well as its role in fostering autonomy and collaboration among students.

Thematic analysis was selected to analyze the qualitative data since it is recognized as a flexible and rigorous method for qualitative research, ensuring trustworthiness when executed systematically (*Kiger & Varpio, 2020*; *Nowell et al., 2017*).

The six phases of thematic analysis, as outlined by *Braun & Clarke (2006)*, provide a structured framework for data analysis, enhancing rigor and facilitating triangulation. Phase 1 involves familiarizing with the data, ensuring immersion and comprehensive understanding, which is foundational for identifying meaningful patterns. Phase 2, generating initial codes, organizes data systematically into manageable units, as seen in categorizing subcategories such as "Engagement" or "Accessibility Issues." Phase 3 consolidates these codes into overarching themes, ensuring alignment with research questions and theoretical constructs. Themes are refined and validated in Phase 4 through reviewing and corroborating with raw data, ensuring coherence. In Phase 5, themes are clearly defined and named, as demonstrated by explicitly naming categories such as

**Table 1 Action research cycles through the lens of activity theory.**

| Phase | Purpose | Action research cycle |
|---|---|---|
| Identifying Students' Needs and Difficulties in Learning English | Needs and difficulties are detected | Reflective diagnosis |
| English class features | | Formulation of hypotheses and interpretative and action objectives |
| | | Literature review |
| Eyeland Design and Students Needs Analysis | An architecture design is planned for the app based on an analysis of the context and students' characterization | Planning a change |
| Teachers' interviews | | |
| Eyeland pedagogical content test | The tasks designed for the app are tested in terms of accessibility, language learning and gamification | Acting and observing the process and consequences of the change |
| Eyeland initial results | The app is described qualitatively in terms of strengths and weaknesses in both the technological and pedagogical field | Reflecting |
| Eyeland presentation | Inform tutors and students about the technical functions of Eyeland app | Replanning |
| Eyeland implementation 1 | Task 1 presentation and execution of tasks phases. Adopt a co-present observation strategy. Take in-room. | Acting and observing again |
| | Research notes. Record video footage of student-presenters' use of Eyeland | |
| Focus groups and interviews | Display students' initial experiences and lesson planning changes | Reflecting |
| Eyeland refinement 1 Lesson planning refinement 2 | Using the feedback to change and modify aspects that were negative | Replanning |
| Eyeland implementation 2 | Task 2 presentation and execution of tasks phases. Adopt a co-present observation strategy. Take in-room. | Acting and observing again |
| | Research notes. Record video footage of student-presenters' use of Eyeland | |
| Usability Survey and interviews | Data collection is analyzed, and initial conclusions emerge about the lesson planning | Reflecting |
| Re-mediation identification through activity theory | Identify re-mediated actions within Eyeland using evaluation frameworks to analyze accessibility, serious games and mobile language learning features | Replanning |
| Eyeland refinement 2 Lesson plan refinement 2 | Establish activity contradictions and tools effect on VIS | Acting and observing again |
| Conclusions about activity model elements | Conclude by summarizing the identified tool-related elements from activity model | Reflecting |

**Note:**
The combination of activity theory and action research has been used successfully in educational research, as proposed by *Orland-Barak & Becher (2011)* and *Behrend (2014)*, and these phases are organized as suggested by theory (source prepared by the authors).

"Cultural or Linguistic Barriers" or "Positive Feedback." Finally, Phase 6 integrates these findings into a coherent narrative, enriched by triangulation from multiple data sources, such as interviews and focus groups. By combining deductive coding frameworks with inductive insights, this structured approach ensures robust, triangulated, and contextually meaningful analysis (see Appendix Thematic Analysis).

These previous data analysis techniques enriched action research cycles that involved teachers in a spiral of self-reflective cycles of the following: planning a change; acting and observing the process and consequences of the change; and reflecting on these processes and consequences; replanning, acting and observing again; and reflecting again (*Kemmis & McTaggart, 2005*). This spiral has a flexible and emergent nature that can be applied in

various ways, allowing for adjustments that are responsive to lived experiences during the research process (*Latorre, 2005*; *Marshall, 2011*; *Nyanjom, 2018*; *Yuni & Urbano, 2006*). Therefore, we agree with *Cohen, Manion & Morrison (2017)* that it is "a powerful tool for change and improvement at the local level" (p. 297).

Table 1 provides the cycles through the lens of activity theory, indicating the iterative process of refining Eyeland's design and pedagogical content based on continuous feedback. It highlights key phases, such as reflection and replanning, which guide modifications to lesson plans and technological features.

## Participants

The participants will include three middle-aged public-school teachers with more than 15 years of experience, four visually impaired students, and eight sighted teenage students from 9th grade and mixed genders who will be observed in three different sessions for tasks 1 and 2. A range of instruments in the different phases of the research will be used: field observations to determine the details of the implementation and lessons through video recording by a co-researcher; focus groups with students at the end of the session to collect impressions, perceptions, and general views of the app; a usability survey to check architecture design feature performance; and interviews with the three teachers before and after the implementation of the app to revise lesson planning re-mediation.

## Data collection

Qualitative research methods and sources allowed for dialog and interaction between the teachers and researchers, as they collaboratively constructed an understanding of how their activity system was changing. The following instruments were selected to collect the information to ensure the elicitation of the components of the activity system and explore the characteristics of the traditional tools and lessons.

### Interviews

Three teachers were asked to participate in interviews as part of a study on the use of the Eyeland app in classrooms. They signed anonymous consent forms, ensuring their identities would remain confidential. While three teachers agreed to use the app and share their experiences, one teacher declined, citing a fear of technology as the reason for not incorporating the app into her lessons. The remaining two teachers participated in interviews conducted over Google Meet, where they shared valuable insights and suggestions with the researchers about the app's impact on classroom engagement and accessibility (see Appendix Transcripts).

### Field observation

Three field observations were conducted to gather data from traditional lessons and assess the initial implementation of the Eyeland app. These observations aimed to identify areas for refinement before the second app implementation and to begin collecting evidence of how the game elements interconnected.

The data collection process involved direct classroom observation, where researchers took detailed field notes and video-recorded sessions to capture interactions between

students and the app. In addition, audio recordings were made of student discussions during the app's usage. The recordings were later transcribed and analysed to identify patterns in student engagement, game interaction, and learning outcomes. Transcriptions of these observations are provided in the appendix.

## Document analysis

Since the aim of the app is remediating the lessons, an analysis of the lesson plans created by the teachers was performed by considering the mediations, themes, language content and different forms of assessment implemented in the recent periods of the courses.

## Focus groups and usability surveys

These two instruments were mostly aligned with the student's experiences while using the app, revealing either negative or positive perceptions and both pedagogical and technological concerns and demands in terms of accessibility, gamification, and language learning. The architecture design and components of the app were also evaluated through questions from both instruments to triangulate the information shared.

## Ethics consideration

To ensure ethical standards, written consent from the participants' parents or guardians was obtained through a Google Forms process before the study. This form outlined the study's objectives, the data collection procedures, and how participant confidentiality would be maintained. Participants' information was anonymized, ensuring no personal data would be shared or published. A copy of the informed consent document is included in the Appendix.

The procedures involved visits from researchers of Universidad del Norte to the participants' classrooms over several weeks, during which time students were given mobile devices in a computer lab setting. The form explained that students would first practice vocabulary and grammar individually and then engage in group missions within the app. These missions included tasks such as locating a place on a map, discovering a lost animal on an island, or searching for a scroll. After completing the missions, students would take a brief quiz to assess what they had learned. The form outlined these steps to ensure parents and guardians were fully informed about the activities their children would be participating in.

In terms of confidentiality, the form assured parents and guardians that their children's participation in the research would be kept confidential. No personal information would be disclosed outside the research team, and each participant's identity would be anonymized by assigning a numerical code instead of using their real names. This measure was put in place to protect the privacy of the participants and ensure that their personal data would not be publicly shared.

The assent form also informed parents that if they chose to allow their child to participate, they would receive a copy of the form for their records. It further emphasized that parents had the right to review the form and consult with the researchers or the Ethics Committee of Universidad del Norte if they had any questions about the study or their

child's participation. Full contact information for the Ethics Committee was provided to facilitate this communication, ensuring transparency and accessibility throughout the research process.

In the case of teachers, they signed anonymous consent forms, ensuring their identities would remain confidential. As it is established by the internal Ethics Committee of Universidad del Norte approved the study (approval 178).

# RESULTS

The findings are organized in an articulated manner with the research-action cycle to answer the questions that originated the study. Equally, these results are consistent with Eyeland activity system are described in the following order to cover all the crucial elements of the theory and establish the points in which they meet.

## Reflective diagnosis: characteristics of traditional English classes

**Research Question 2: What are the characteristics of traditional EFL lessons designed for both sighted and visually impaired students?**

The reflective diagnosis revealed several through the interviews conducted key characteristics of traditional English lessons. These classes were largely grammar-based and focused on teacher-centered instruction, with limited opportunities for communicative skill development. Tools used in these lessons were primarily visual and heavily reliant on printed materials and teacher input. For VISs, magnified materials, Braille, audiobooks, and recorders were the primary aids provided. However, these tools were insufficient to meet the needs of students with visual impairments. Teachers identified weaknesses in gamification, accessibility, and mobile language learning, noting a lack of sufficient digital materials or devices tailored to visually impaired students. The use of Braille materials provided by the government was also found to be inadequate. This reflective analysis helped to identify the barriers hindering inclusive education in traditional lessons, particularly for students with blindness or low vision.

Table 2 presents an overview of the organization and equivalence of elements from activity theory, including how subjects, tools, and other components interact in various contexts such as interviews, observations, and focus groups.

The analysis of this reality is deepened based on the documentary analysis and observations 2 and 3. This made it easier to compare traditional media and Eyeland emerged. From this information, specific accessible functionalities are outlined as depicted in Table 3.

One of the initial observations concerned the traditional lessons' grammar-based focus, which often failed to match students' actual proficiency levels or promote communicative skills effectively. The tools in these lessons were primarily visual and printed, heavily reliant on teacher input and board work. This mismatch between the traditional lesson structure and students' communicative needs presented a contradiction: the desire to enhance student engagement with new digital tools like Eyeland *versus* the deeply entrenched reliance on traditional methods. While traditional mediation tools like

**Table 2 Findings on organization and equivalence with activity theory elements.**

| | |
|---|---|
| Interview 1 | Community of practice |
| Observation 1 | Subject + tool |
| Document analysis | Rules |
| Observation 2 | Subject + tool |
| Usability survey | Division of labor + tool |
| Focus groups | Subject + tool |
| Observation 3 | Subject + tool + object |
| Interview 2 | Community of practice |

**Note:**
This chart outlines the different data collection methods used in a study and the corresponding components of the activity theory framework they are associated with.

**Table 3 Comparison of traditional and new mediation.**

| Traditional mediations | New mediation |
|---|---|
| Magnified materials, Braille, Audio Books Recorders, Special needs assistant, Realia | Eyeland App |
| **Accessibility** | **Accessibility** |
| Audiobooks were able to be manipulated by VIS | 8 Accessibility buttons, Screen reader, Superpower activation (super hearing: pronunciation anytime) |
| **Gamification** | **Gamification** |
| No presence | Serious game |
| Mobile learning | Mobile learning |
| No presence Slate | Individual and collaborative roles |

**Note:**
Traditional mediations for visually impaired students (VISs) are compared to the newly introduced mediation, "Eyeland," an accessible serious game designed for learning English. This study provides detailed insights into the incorporation of gamification and mobile learning elements, which were previously absent in traditional methods.

magnified materials, braille, audiobooks, and recorders were in place for visually impaired students, the new mediation *via* Eyeland introduced a task-based, game-like environment that integrated these features and added more accessibility.

Through the first observation of a lesson mediated by Eyeland, it was noted that students with visual impairment were working in the initial stage individually since the task demanded becoming familiar with the vocabulary and functional grammar. The teachers play a different role here since they are facilitating and monitoring progress instead of presenting the content themselves. The accessible content expanded the opportunities for interaction and authentic input for the students.

In this shift, tensions emerged around the roles of teachers, who transitioned from being content providers to facilitators monitoring student progress. This change reflected a broader contradiction between teacher-centered instruction and the more student-centered approach enabled by the app, particularly as the accessible content, fostered greater interaction and authentic input.

## Construction of the action plan

**Research Question 1: How can an EFL-accessible task-based serious game app (Eyeland) remediate traditional lessons for both sighted and visually impaired students at a public high school in Colombia?**

The action plan involved introducing the Eyeland app as a task-based, serious game designed to improve accessibility for both sighted and VISs. The app sought to overcome the limitations of traditional lessons by offering a more engaging, interactive, and personalized learning experience. It integrated features like gamification and task-based learning, which were identified as gaps in traditional teaching. The design of the app focused on providing more inclusive learning experiences, utilizing digital tools and accessibility features such as voice navigation, audio instructions, and gamified tasks to support them.

In terms of division of labor, creating accessible serious games like Eyeland requires understanding the different categories of disabilities and their impact on game use. This is a challenging endeavor, as incorporating accessibility guidelines into any game design requires considerable effort. However, the results demonstrated that the gaming dimension of Eyeland facilitated more engaging, student-centered learning experiences. This raised a tension between the effort required to create accessible games and the benefits they provide in making learning more enjoyable and inclusive. The game aspect supported core pedagogical principles, enhancing the learning experience for all students, but it also required a shift in how teachers and students traditionally operated in the classroom.

## Transformative action

**Research Question 3: What adjustments were made to the EFL lessons while the Eyeland app was used for both sighted and visually impaired students?**

The introduction of the Eyeland app brought significant changes to traditional EFL lessons. The app allowed for a shift from teacher-centered to student-centered learning. Lessons became more interactive and task-based, with students participating in activities that required them to collaborate in groups, record voice notes, and engage in gamified exercises. Classroom layouts were rearranged to facilitate group work, and students were given more autonomy in navigating their learning tasks. This re-mediation through Eyeland provided a more dynamic and inclusive learning environment, particularly for VISs, who reported increased motivation and engagement.

These changes allowed for new forms of participation and interaction, addressing contradictions between traditional passive learning and the active engagement promoted by the app. The new lesson format 5E model allowed to outline the following aspects:

**Engagement**: The lesson begins by generating student interest through gamified tasks introduced by the Eyeland app. This phase allows the teacher to assess students' prior knowledge while introducing new content in an interactive way.

**Exploration**: Students participate in hands-on activities and collaborative group work.

**Explanation**: During this phase, students articulate their understanding of the material as they interact with peers and the app's features. Teachers facilitate explanations by encouraging students to discuss and reflect on their experiences with the app.

**Elaboration**: Students apply their learning in new, meaningful contexts provided by Eyeland's accessible tools. For instance, the incorporation of endemic animals and places from real world sceneario.

**Evaluation**: The evaluation phase includes assessments that measure students' progress at a formative level, allowing repetition of exercises and error corrections.

## Reflection, interpretation, integration, and replanning

**Research Question 4: What activity re-mediation occurred while using Eyeland?**

The use of Eyeland transformed the traditional division of labor within the classroom. Teachers shifted from being content providers to facilitators, monitoring student progress and fostering autonomous learning. The app supported greater interactivity and authentic input, which allowed students to engage more deeply with the learning material. The integration of gamification elements made learning more enjoyable, particularly for students with disabilities. However, the transition to this new method of teaching presented tensions, particularly for teachers who were more accustomed to direct instruction. Some teachers struggled to fully embrace the more flexible, student-controlled learning environment promoted by the app.

In terms of curriculum alignment, the app helped address the mismatch between the curriculum's rigid grade-based structure and the students' actual language proficiency levels. While traditional lessons placed students according to their grade level, Eyeland allowed for more personalized learning paths that better reflected individual student needs, particularly in terms of grammar and communicative skills.

## Evaluation of learning outcomes and student experiences

**Research Question 5: What are the experiences of sighted and visually impaired students using Eyeland?**

As part of the study, pre- and post English tests were applied to the group of students, to measure the impact of the use of the app. Figure 5 shows the results for the twelve students that were evaluated. Four of the students with low vision, and eight students did not have any vision problems. The test had five categories: speaking (s), verbal (v), listening (l), reading (r), and writing (w). For both groups it is clear that they saw an improvement for each of the skills, after the use of the application. The only category where the improvement was not high was for reading. For the students with no visual issues, the improvement was from 5.5 to 6.25 on average, while for the other group, there was no significant difference. For the other categories however, the improvement was significant, and the highest difference was observed in the speaking category.

As demonstrated in Fig. 5, the contrast between the VISs and sighted students progress is outlined per language skills categories.

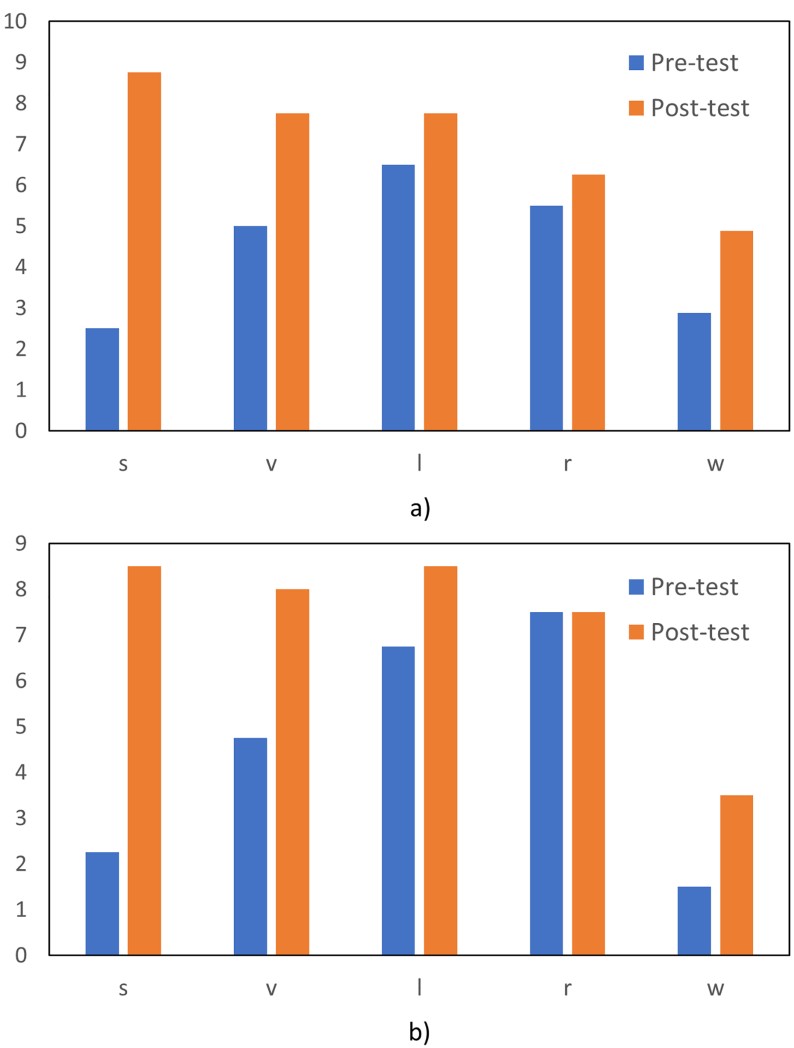

**Figure 5 (A and B) Pre-test and post-test results.** Both groups showed significant improvements in their post-test scores, with the VISs achieving higher overall scores, as illustrated in the lower graph, while the sighted learners' results are presented in the upper graph.

From the transcripts of the app-enabled lesson, we observed how the students engaged in the use of the application, highlighting the fact that it used gamification to help the students stay focused and interested in continuing the activity. The following are some comments: From a student with low vision (proudly): "I'm ready to use my Super Hearing powers! Shhh let me listen." Another student with no vision problems commented: "I want to be the Memory Pro! I have a great memory for vocabulary." Another student, selected as leader of its group, commented: "Our group is working well, but the definitions are a bit tricky." The comments show an interest from the students, who felt highly interested, but sometimes showed the tasks were not easy and fast to complete, and needed team work to complete them correctly. More comments and the complete transcripts of the app-enabled class are annexed to this work.

## DISCUSSION

EFL traditional lessons designed for both sighted and visually impaired students in Colombian public schools face notable challenges due to limited resources. Conventional teaching methodologies predominantly rely on visual and printed materials, which may curtail the inclusivity of lessons, especially for VISs. These traditional pedagogical paradigms tend to emphasize grammar-based isolated topics, further limiting the accessibility and effectiveness of EFL instruction for both sighted and VIS learners (*Lin & Yunus, 2012*; *Karamifar et al., 2019*; *Durham, 2022*).

One major constraint identified in implementing the app was the lack of teacher participation due to attitudes toward technology use in the classroom and resource limitations. Some teachers felt disconnected or frustrated by their lack of knowledge and interest in new technologies, despite recognizing their potential benefits. This aligns with the literature indicating that while educators acknowledge the role of ICTs in lesson design and language acquisition, they often cling to traditional methods (*Lin & Yunus, 2012*; *Karamifar et al., 2019*; *Durham, 2022*). Teachers' reluctance to adopt ICTs exacerbates the digital divide, particularly in vulnerable schools. Future research should explore teachers' knowledge, perceptions, and ICT skills to better integrate these technologies (*Ayub, Bakar & Ismail, 2015*; *D'Luyz Monsalve, 2021*; *Tapia Silva, 2018*). In both implementation sessions of Eyeland, the teachers showed little interest in how the app might assist them with lessons. In the Colombian context, apathy, beliefs, and misconceptions about inclusion often lead to unclear strategies and wrong priorities within educational culture. Despite this, the teacher's role as the central educational agent remains crucial. Teachers must develop positive attitudes to help students feel that they are part of a family and community (*González-Rojas & Triana-Fierro, 2018*; *Medina García, 2016*; *Prieto, 2008*).

Adjustments to the EFL lessons *via* the Eyeland app treated the app as an assistive technology mediator for VIS, necessitating a paradigm shift in lesson planning away from conventional approaches. This integration required novel instructional strategies and aligned with the 5E instructional model by *Tucker (2020)*, with a special focus on engagement, as the app served as a warm-up and collaborative tool. *Duckett & Pratt (2001)* underline the importance of information technologies for VIS, whereas *Fraser & Maguvhe (2008)* and *Söderström & Ytterhus (2010)* emphasize the use of technology and collaborative learning strategies to address common challenges. The 5E model's evaluation element focuses on interaction, guiding mobile learning design to integrate apps with other classroom resources and improving student outcomes (*Rozario, Ortlieb & Rennie, 2016*). Action research findings suggest that teacher education programs must develop skills in aligning lesson plan features and using assessment data to better understand students' learning needs, moving beyond traditional forms of assessment (*Chizhik & Chizhik, 2018*).

The activity re-mediated by Eyeland focused significantly on accessibility. Traditional mediation tools such as audiobooks are limited in effectiveness for VIS, offering no topic-related gamification activities. In contrast, Eyeland allows VISs to experience both individual and collaborative EFL learning, utilizing features such as a friendly screen reader with eight accessibility buttons and interactive games that include super activation,

superheating, and pronunciation. This aligns with prior studies indicating that current gamification designs often emphasize navigation, orientation, and mobility training but lack collaborative interaction (*Tan, Huang & Shang, 2022*). The interaction design for VISs usually includes sensory augmentation and substitution through spatial audio and tactile interfaces, raising questions about the complementary of information from different modalities. The app's superpower features encourage researchers to explore personalized dynamic adjustments of content and tasks, aiming to establish standardized methods for evaluating effectiveness.

The primary contributions of this research are that an EFL-accessible task-based serious game app can remediate traditional lessons for both sight and VIS by integrating principles derived from ATMSG. This model shifts from treating the game as an isolated entity to recognizing its pivotal role within a dynamic system involving learners and educators, enhancing overall classroom accessibility. The sighted and visually impaired students' experiences using Eyeland were predominantly positive. The app increased motivation and engagement and improved attitudes toward speaking and listening skills. The game's personalized levels and active player roles significantly enhanced motivation and learning. Accessibility features, such as a friendly screen reader and interactive games, greatly improved the learning experience for VIS. Each level of the game is tailored to the player's abilities, engaging players and improving motivation in learning (*Michael & Chen, 2005*; *Oblinger, 2004*; *Lepper & Malone, 2021*).

However, despite the predominantly positive feedback, some challenges were highlighted during the interviews and focus group discussions. One of the key negative aspects identified was the persistent resistance of teachers toward adopting new technologies, although there was growing awareness of the potential benefits of these tools. Teachers often expressed frustrations with technological tools due to limited training and a lack of familiarity, which hindered the full integration of Eyeland in their lesson plans. Another issue raised by participants was the interaction design of the app, particularly in terms of the exercises and word manipulation tasks. Suggestions were made to improve the interactivity by allowing students to drag and manipulate words more easily. Additionally, some VISs noted that reducing exterior noise during gameplay would significantly enhance their ability to focus and understand the app's instructions, improving the overall learning experience.

This study confirms the potential of activity theory to play a vital role in advancing the field of mobile learning by providing valuable insights and tools. It serves as a valuable lens for analyzing learning processes and outcomes, guiding the design of effective mobile learning experiences. It offers a framework for designing context-aware applications, which are fundamental for the success of mobile technologies. Activity theory proposes a model that elucidates the relationships between different environmental variables, addressing key factors that influence human engagement in mobile applications (*Kaenampornpan & O'Neill, 2004*; *Uden, 2007*). The study's central findings highlight substantial transformations in lessons due to the integration of accessibility features and multimodality, specifically through Eyeland's incorporation of the 5E model for lesson

planning. VIS gave overwhelmingly positive feedback, indicating the application's effectiveness.

## CONCLUSION

The study suggests the potential emergence of innovative activity systems within serious game contexts, reinforcing inclusive classrooms by balancing accessibility, gamification, and mobile language learning. A noticeable research gap exists in the exploration of the integration of serious game applications within the EFL domain. Therefore, ATMSG has proven valuable for researchers seeking to understand the intricate dynamics of gaming environments and their influence on pedagogical planning.

Future research should prioritize examining ways to facilitate language learning for visually impaired students (VISs), recognizing it as a fundamental right to education (*Kocyigit & Artar, 2015*). Remediating and redesigning educational scenarios are crucial to ensure inclusivity. This involves incorporating accessible tools and methodologies that provide equal learning opportunities for VIS.

Another recommendation is to advance mobile learning for VIS through activity theory since it plays a vital role by providing insights and tools for designing effective learning experiences. It offers a framework for context-aware applications, which is essential for the success of mobile technologies. Educators should expand the use of such technologies in educational institutions to enhance the learning process and outcomes.

The integration of Eyeland as an EFL-accessible task-based serious game app has significant potential in remediating traditional lessons for both sighted and visually impaired students. The app facilitates the adaptation of lessons through a dynamic system involving both learners and educators, enhancing overall classroom accessibility. Despite challenges in teacher participation due to attitudes toward technology, the positive experiences of students highlight the transformative potential of Eyeland in promoting inclusive and effective language learning environments.

It is crucial to acknowledge that language learning for VISs is not a discretionary matter but a fundamental right to education, inseparable from broader human rights considerations (*Kocyigit & Artar, 2015*). Therefore, remediating and redesigning these educational scenarios is imperative.

The combination of a meticulously structured lesson plan with an easily accessible serious game application shows promise in cultivating favorable educational experiences for both sighted and visually impaired individuals. This approach shifts from viewing serious game applications as standalone tools to leveraging the collaborative synergy of system components, broadening the horizons for successful learning outcomes. Artificial intelligence (AI) powered voice assistants, smart readers, and natural language simulations can enhance speaking and listening skills, ensuring accessibility and contributing to inclusive, effective language learning environments.

In essence, addressing challenges in teacher participation and technology integration is critical to harnessing technology's full potential in creating inclusive and effective learning experiences. Ongoing teacher education programs should focus on enhancing digital literacy and fostering inclusive lesson planning and assessment (*Chizhik & Chizhik, 2018*).

By improving teachers' knowledge, perceptions, and skills in using ICTs, educators can better integrate these technologies into their teaching practices, ultimately benefiting all students.

In conclusion, while the study's findings are based on a specific cohort of visually impaired students in face-to-face environments, they may not fully represent the broader population. Further studies should seek to include diverse educational populations and explore AI-powered tools or components to assist visually impaired students, aiming to continuously improve this area. By addressing these limitations and strategically targeting areas for enhancement, subsequent research can significantly improve the overall educational experience for visually impaired students, promoting a more inclusive and equitable learning environment.

### Funding

This work was supported by EPICS funding in IEEE from Ashley Moran. The funders had no role in study design, data collection and analysis, decision to publish, or preparation of the manuscript.

### Grant Disclosures

The following grant information was disclosed by the authors:
Engineering Projects in Community Service (EPICS) in IEEE.

### Competing Interests

The authors declare that they have no competing interests.

### Author Contributions

- Karen Villalba conceived and designed the experiments, performed the experiments, analyzed the data, prepared figures and/or tables, authored or reviewed drafts of the article, and approved the final draft.
- Heydy Robles conceived and designed the experiments, performed the experiments, prepared figures and/or tables, authored or reviewed drafts of the article, and approved the final draft.
- Miguel Jimeno conceived and designed the experiments, performed the experiments, performed the computation work, prepared figures and/or tables, authored or reviewed drafts of the article, and approved the final draft.
- Martha Cecilia Delgado-Cañas conceived and designed the experiments, performed the experiments, analyzed the data, authored or reviewed drafts of the article, and approved the final draft.
- Adriana Perez conceived and designed the experiments, prepared figures and/or tables, authored or reviewed drafts of the article, and approved the final draft.
- Francisco Quintero conceived and designed the experiments, analyzed the data, authored or reviewed drafts of the article, tested the app since he is blind, and approved the final draft.

## Ethics

The following information was supplied relating to ethical approvals (*i.e.*, approving body and any reference numbers):

The internal Ethics Committee of Universidad del Norte approved the study (178)

## Data Availability

The raw data are available in the Supplemental File.

## Supplemental Information

Supplemental information for this article can be found online at http://dx.doi.org/10.7717/peerj-cs.2631#supplemental-information.

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
