# Peer review of "An activity theory-based exploration of “Eyeland”, a task-based serious game for EFL visually impaired students"

_PeerJ Computer Science, doi:10.7717/peerj-cs.2631_

## Round 0.1 · original submission · Major Revisions

Dear authors,

You are advised to critically respond to all comments point by point when preparing a new version of the manuscript and while preparing for the rebuttal letter. Please address all comments/suggestions provided by reviewers, considering that they should be added to the new version of the manuscript.

Kind regards,
PCoelho

Reviewer 1 ·

Basic reporting

1. The language of the manuscript needs professional editing. Some parts are unclear or not structurally correct.

2. Literature review lacks most recents studies from the past 3 years, 2021 onwards, so there should be added studies to the review. Following the review you should add a paragraph to highlight the gap in the literature considering recent studies.

3. Some statements are unclear, for example, in field observation; “These observations were recorded and transcribed to be presented in the appendix”. Here it is necessary for the authors to state the procedure for data collection, how were they recorded? There is no mention of particpants consent during the study. By this sentence do you mean the transcriptions are in the appendix? Where are the results for the observations recorded?

The Division of labor subheading is not appropriate, you might want to consider moving this information to another part above.

Subheadings “Subject” , “Object”, ?? There are too many subheading that disturb the flow and coherence of the text. A major revision is necessary in the organization of the paper.

Experimental design

4. The authors very briefly touch upon the data collection process, which needs further elaboration. There are many steps in the data collection process, the interviews, observations, lesson plan analysis, focus group discussions, surveys, which need to be clarified and explained in this section.

5. Consent was obtained from the young learners, what about the teachers consent forms?

6. The highlighted sentence in the text under ethics data analysis needs revision.

7. Can you elaborate what you mean by ethics data analysis? What is analyzed in this section?

8. There is a mention of only one material provided for visually impaired by the public schools, but not stated what. You need to be specific with details.

9. In line 391 , “sighted” is used. Is this partially sighted? The technical vocabulary used should be revised.

10. Line 396, what is a research sub question? The terminology used is at times unclear.

11. There is serious issue with terminology used, as there needs to be consistency across the manuscript, professors-teachers-instructors?… be consistent

12. Line 414-415, two things are compared which are not clear, topics are compared with verbs tense…. What is the basis for this comparison… it leaves the reader confused.

Validity of the findings

13. The results section does not provide enough information of the data analysis and the outcomes. Where are the results of interviews and focus group discussions? The results section lacks details proofs (maybe consider including tables & diagrams to show the results to help the reader get a general picture of the outcome)

14. The manuscript lacks visual depiction of the results, which is necessary for this type of study, both in the procedure section and the results section

15. Research questions need to be modified, its better to have them numbers, not as sub questions.

16. Again specifiy what you mean by sighted?

17. Methods section is not described in details with information provided to allow replication of the study, therefore it is necessary to add these details.

18. Findings and results section need a major revision to include details about each of the sections for qualitative data analyses.

19. Results of interviews and focus group discussion are not clearly stated and its relevance to the discussion needs to be elabored in the discussion section. In the discussion section you mention the results to be predominantly positive, here you need to also report on the negative aspects.

20. Since the study framework is based on MALL (Mobile assisted Language Learning) there should be a section added to the literature to elaborate on this

Additional comments

The topic of this research of great significance and deserves publication; however, major revisions are necessary to achieve publication standards.

Annotated reviews are not available for download in order to protect the identity of reviewers who chose to remain anonymous.

·

Basic reporting

Dear Authors,

I have critically examined your manuscript. Your use of activity theory to explore a task-based serious game for EFL visually impaired students is intriguing, and I believe there is limited research in this area. However, the method's biggest flaw appears to be the lack of a specific intervention, which the research title emphasizes. Specifically, the results of my examination of your article are presented below:

1. The abstract has covered most of the necessary parts. In your revision, please include the data analysis techniques you employed for each type of data used to answer the research questions. Also, the first and second sentences are not coherent; you need to rewrite them into one sentence and build close connections in order to scaffold a distinctive context for your research.

2. The whole introduction is not coherent. For example, the first and second paragraphs are not well connected. You abruptly introduced the terms VIS and MALL in the second paragraph, despite not mentioning them in the first. Furthermore, the first paragraph lacks clarity and fails to specifically address the focus of your research, which is clearly indicated in the title. You should immediately discuss the development of the research area in the context of activity theory, task-based serious games, and visually impaired students. Following this, you can succinctly outline the areas that previous studies have delved into and those that they haven't, thereby highlighting the contribution your study seeks and its uniqueness. Moreover, you should avoid writing brief paragraphs and presenting multiple references without emphasizing the context-specific research contexts and findings of the cited references.

3. The literature review section lacks criticality and coherence. Again, you should avoid writing brief paragraphs and presenting multiple references without emphasizing the context-specific research contexts and findings of the cited references. You need to substantially revise this section, making the reviews more critical, thorough, and contextual. Additionally, demonstrate how the context of your study benefits from, informs, or utilizes these reviewed findings.

Experimental design

In terms of research design, the method section is weak. Present your study as though it took place after the introduction of task-based serious games into EFL classes. In this case, you will need to provide detailed information about the course design and learning activities. Currently, the research procedures lack solid foundations. Simply put, the abruptness of data collection appears to undermine the strength of your presentation. While I believe that you did implement the treatment, these details are missing or unclear in the article. Furthermore, the explanation of the procedures for ensuring reliability and validity, as well as the details of the data analysis techniques, is inadequate. This section requires significant revisions.

Validity of the findings

For qualitative research, the presentation of the qualitative findings is not convincing and does not adhere to standard reporting practices for qualitative data analysis. Basic elements, such as participant excerpts, are missing. Reporting qualitative findings without these excerpts can make them appear as personal opinions rather than evidence-based insights. Additionally, the reported qualitative insights lack critical analysis. However, the discussion section seems sufficiently developed.

Additional comments

-

---

## Round 0.2 · Minor Revisions

Dear authors,

Thanks a lot for your efforts to improve the manuscript.

Nevertheless, some concerns are still remaining that need to be addressed.

Like before, you are advised to critically respond to the remaining comments point by point when preparing a new version of the manuscript and while preparing for the rebuttal letter.

Kind regards,
PCoelho

Reviewer 1 ·

Basic reporting

Very good job of making revisions to the structure and clarity of the manuscript.

Experimental design

The authors have made a good attempt to revise and add missing details to the methods section. Well done. Revision of the research questions has also made them clear and understandable.

One minor revision is still necessary in the Qualitative data analysis section, as explained below.

Validity of the findings

Well done in the revision. Findings are linked to the original research questions and discussion adequately supports the study rationale.

Additional comments

One important note in the data analysis of the qualitative phase. It has not been specified what qualitative analysis technique has been used...
The authors stated "Qualitative data from observations, focus groups, and interviews were transcribed, with a focus on identifying patterns in student engagement, accessibility issues, and the impact of the app on teaching practices. The data analysis process followed the steps of familiarization, coding, and generating themes to ensure a rigorous exploration of the findings." => this does not inform the reader of the analysis technique used. What common themes were generated from the interviews, focus group discussions, observations, etc. How was coding and generating themes triangulated with the previous data? These issues still need to be addressed.

One more issue: Literature review is repeated as a heading twice.

---

## Round 0.3 · accepted · Accept

Dear authors, we are pleased to verify that you meet the reviewer's valuable feedback to improve your research.

Thank you for considering PeerJ Computer Science and submitting your work.

Reviewer 1 ·

Basic reporting

The authors have done a good job at revising the manuscript

Experimental design

Well done

Validity of the findings

Revision well done based on all comments provided

Additional comments

The paper is well revised and comments are adequately addressed. Manuscript is now ready for publication.